# Reconceptualizing Activism through a Feminist Care Ethics in the Ontario (Canada) Early Childhood Education Context: Enacting Caring Activism

Brooke Richardson [1,*], Alana Powell [2], Lisa Johnston [3] and Rachel Langford [4]

1   Department of Sociology, Brock University, St. Catharines, ON L2S 3A1, Canada
2   Association of Early Childhood Educators of Ontario, Toronto, ON M6G 1A5, Canada
3   Faculty of Education, York University, North York, ON M3J 1P3, Canada
4   School of Early Childhood Studies, Toronto Metropolitan University, Toronto, ON M5B 2K3, Canada
*   Correspondence: brichardson@brocku.ca

**Abstract:** While early childhood education (ECE) in Ontario has always had a vibrant social activist community, it is characterized by tensions within and between individuals and institutions at the minor (childcare centres, post-secondary ECE programs) and major (mainstream media, public policy) levels. ECE activism is further complicated by the fact that it often feels impossible/unsustainable within our existing patriarchal, neoliberal political structure. In this paper we, four ECE activists and leaders, turn to feminist care ethics (FCE) to reflect on our own activism experiences and imagine a different way of doing and sustaining activism in ECE. We insist that activism be understood as a relational *process* that bridges major and minor spaces (and everything in between) in a way that cares *about*, *for*, and *with* all those involved. We enthusiastically invite other to join us on this journey, exploring and navigating the beautiful awkwardness, discomfort, tension, and possibilities in caring *for* and *with* each other in major and minor political spaces.

**Keywords:** feminist care ethics; activism; advocacy; early childhood education; childcare

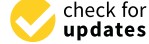



## 1. Introduction

"Bringing the question of woman into the philosophical canon essentially deconstructs that cannon. Women don't fit". —Hekman (2014, p. 24)

This paper is about *if* and/or *how* early childhood education (ECE) activists in Ontario (Canada) can reshape, if not deconstruct, the hegemonic, patriarchal cannon in which we (alongside children and families) exist. ECE has always had a vibrant social activist community, despite the lack of public policy infrastructure necessary to meet the care needs of young children, their families, educators, and communities. The not-for-profit, province-based (but not government funded) professional association for early childhood educators (ECEs) and grassroots advocacy association, alongside several other smaller student and community groups organizing at the minor (ECE programs) and major (policy) level, have worked tirelessly over the past decade to build strength and solidarity in an increasingly market-driven childcare context. Motivated by the desire to create a world in which all can live as well as possible in the increasingly impossible/unsustainable social, political, and environmental contexts this special issue highlights, the authors are or have been active members of this activist community over the past decade (or longer).

While attempting to work towards childcare policies that meaningfully engage with the intersecting needs of children, families, and educators (Richardson et al. 2021), we have also observed and experienced tension, stress, strain, and conflict. Strongly held ideas, values, and practices intersect within and between the major and minor political places, too often fracturing, rather than strengthening, the broader goal of pursing an ECE system that meets the needs of children, families, and educators. Activism efforts are also complicated

by a (necessary?) take up of a politics of refusal—or "the art of voluntary inservitude" (Ball 2016, p. 1136) at the minor level (this is discussed further below).

As activists, we internally and collectively struggle with how to politically strategize without compromising our own values, beliefs, and desires. For example, there has been an ongoing tension in ECE advocacy in relation to the human capital/social investment rationale, as this way of thinking (public investment justified through immediate and long-term economic returns on investment) "fits" with neoliberalism (Prentice 2009; Langford et al. 2013). We question if, or to what extent, we should shapeshift (draw on this discourse even though it may not be what ultimately motivates us) to be taken seriously. Yet, we fear such shapeshifting takes us further away from ourselves and each other, thereby creating more tension within and between individuals and institutions while further trapping us in existing political structures and ideologies (Hekman's proverbial "cannon").

Reflecting on some of our own activism experiences, we wonder if and/or how reconceptualizing activism through a feminist care ethics may create inroads to new political ways of being—particularly ways that are both authentic and work to destabilize the binaries of major and minor political spaces.

### 1.1. Our Positioning

We wish to acknowledge our privileged positions and relations within the ECE sector in Ontario. Our experiences are inherently limited by the fact that we are four middle-class White women, all in positions of leadership within the ECE sector. While we all identify as early childhood educators (ECEs) and have practiced as ECEs at some point in our careers, none of us are currently professionally working with children and families in our daily lives. Instead, we occupy positions of leadership within the academy (contract lecturer, Professor Emerita, and graduate student) and the ECE community (leader of a community of practice, Executive Director of an ECE organization, and Past Presidents of the Association of Early Childhood Educators of Ontario).

We self-identify as activists—a carefully chosen word which we all feel is central to our personal identities. We choose the word "activist" and "activism" intentionally as it emphasizes acting *with* others (as opposed advocating *for* others) in an ever-evolving, non-linear, relational process we hope will lead to a more caring world. We use the word "activism" in a way consistent with Kahn and Lynch-Biniek's (2022) thinking about "organizing": "aggregating people so they can strategically weird their combined strength") (p. 330).

All four authors are deeply influenced by a feminist care ethics, whereby we highlight the complex, power-laden, and ethical nature of caring relations at every level (Sevenhuijsen 1998; Tronto 2013, 2016). In our attempts to carry out our work with care, we make a concerted effort to ensure we remain open to new ways of thinking, understanding, and being in an ever-changing world. We actively work to seek out voices in our activism work that extend beyond our own—particularly Black, Indigenous, and other racialized communities. We recognize we are not perfect in these attempts and that our own privilege and bias undoubtedly cloud our work. In this way, we welcome others to join us in our ongoing journey of reconceptualizing activism in ECE.

### 1.2. The Ontario Childcare Context

It is important to recognize that Ontario is politically located in the decentralized federation of Canada. Childcare falls under provincial jurisdiction whereby curriculum, health, and safety regulations, qualifications/education of staff, and funding models are determined. At the time of writing, a market, fee-for-service model of childcare dominates the country[1]. Generally speaking, there are not enough regulated childcare spaces for children/families who need them and market-determined parent fees are unaffordable for most families. Both of these chronic problems became even more pronounced during the COVID-19 pandemic as childcare centres struggled to remain operational amidst prolonged uncertainty (Macdonald and Friendly 2021).

There is a silver lining, however. Arguably because of the highly visible and mediatized (though both social and mainstream media) gendered impact of the pandemic, the federal Canadian government is now taking an unprecedented level of leadership in funding a national childcare system (Richardson et al. Forthcoming). The federal government has promised upwards of CAD 30 billion over the next decade to achieve a CAD 10/day childcare system across the country. While childcare has been positioned as a national policy priority by childcare activists, researchers, and the community for decades, this is the first time it has been taken up by legislation that commits the federal government to bringing a national system to fruition. As of March 2022, each province and territory signed asymmetrical bilateral agreements with the federal government outlining the funding parameters of federal monies. While commitments to various aspects of childcare differ in each province (particularly in relation to workforce issues and expansion plans), there has been an overall shift towards repositioning childcare as a national policy issue. In this paper, we are particularly focused on Ontario—the last province to sign onto the national childcare program. Since the provincial Conservative government was elected in June of 2018, accessible, affordable, high quality childcare programs with well-educated and decently remunerated staff have not been, and are still not, a priority. It is within the Ontario context that our reflections on our own activism and reconceptualizing a caring activism take place.

### 1.3. Defining Major and Minor Political Spaces

Before reviewing key literature that informs our work, it is necessary to define what we mean by "major" and "minor" spaces, as we rely on this language throughout the paper. We are thinking about major political spaces as those that are public facing, conflictual, and populated by individuals with policy/political influence (this may or may not include formal government representatives). Major political spaces can take the form of a policy task force, consultations with government, engaging in public advocacy campaigns, and/or interacting with the mass media. Major political spaces are often high stakes and intimidating as one may visibly commit to ideas or positions that may not be popular with government(s), childcare program administrators, the general public or even colleagues. It is important to recognize that these spaces are often exclusionary to those without "power". In contrast, minor ECE spaces are understood to be more private, typically populated by those with similar ideas and values (or at least an appreciation of different values/goals/ideas) and are more accessible. Childcare centres, postsecondary ECE programs, community groups within the sector, and professional learning organizations primarily function at the minor level. Minor ECE spaces commonly might be thought to exist outside the public world of politics.

Along with our colleagues, and particularly through a feminist care ethics (FCE) lens, we understand all ECE spaces to be deeply political and ethical (Dahlberg and Moss 2005; Moss 2006, 2007; Sevenhuijsen 1998), non-hierarchical (neither is more or less important than the other), and inextricably interconnected. Almost twenty years ago, Dahlberg and Moss (2005) gestured towards the fluidity of major and minor political spaces when they said: "Major politics may create conditions for minor politics, minor politics may, 'connect up with a whole series of other circuits' [Rose 1999] and with such force as to disturb and direct major politics" (Dahlberg and Moss 2005, p. 138). These authors go onto explain that "politics does not involve an opposition between minor politics and 'a traditional or majoritarian' politics in the sense that a choice has to be made. A democratic society needs both" (Dahlberg and Moss 2005, pp. 137–38). While activism is traditionally thought to occur only at the major level, we insist that engaging within and *across* major and minor political spaces is not only possible, but *necessary*, for meaningful change to occur. Thinking with FCE, we further insist that activism be understood as a relational *process* that bridges major and minor spaces (and everything in between) in a way that cares *about*, *for*, and *with* all those involved. In other words, it is only through responsive, caring relationships

between individuals and groups within and across major and minor political spaces that the difficult, complex, relentless, and transformative activism work can be done/sustained.

A Politics of Refusal

In contrast to this thinking, ECE scholars have questioned whether dedicating copious amounts of time and energy to major politics was worthwhile, as deeply entrenched neoliberal values and beliefs emphasizing efficiency, accountability, and individual responsibility felt impenetrable. This is an important consideration, especially when there are already major barriers to shifting potential ECE activists from the "sympathy pool" to the "participation pool" (Almeida 2019). For example, Moss (2017) asked whether macro-level activism in ECE is "a forlorn case of spitting in the wind" (p. 27). While he went on to assert that "policy matters", he suggested that engaging with minor politics may offer the most fruitful use of limited time and energy of ECE stakeholders until hegemonic, neoliberal discourses subside (p. 27). Reflecting this sentiment, Pacini-Ketchabaw et al. (2015) posited that working with neoliberally entrenched governments limits, rather than expands, possibilities for the ECEC sector.

Government decision-making and actions are based on traditions, values, and rationalities prescribed by dominant discourses and ideologies (Dahlberg and Moss 2005). Minor politics, on the other hand, are small spaces where people negotiate power/knowledge relations, consider alternate discourses, and think about creative possibilities (p. 182).

Moss (2017) proposed a "politics of refusal" as one way for early childhood educators and allies to decide where to focus their energy. These authors quoted Stephen Ball's (2016) definition of the "politics of refusal" as: "care of the self . . . a continuous process of introspection which is at the same time attuned to a critique of the outside world . . . . this is the art of voluntary inservitude, of reflective indocility" (p. 1136). While we agree with these authors that such introspection at the minor level (including the "refusal" to take up dominant oppressive neoliberal discourses) is necessary, we are concerned about what happens (or does not) if/when one refuses to engage with/in major political spaces entirely. As researchers/academics in positions of privilege, we cannot see an ethical way to refuse engagement in major political spaces when hegemonic ideas and practices continue to devalue our own and our colleague's highly gendered work, knowledges, and experiences (see Kahn and Lynch-Biniek 2022 for a more thorough discussion of the complexities of activism/organizing in the academy). We contend that transformative change will occur only with persistent, conscientious attempts to disrupt the neoliberal story at *both* the minor and major political level.

At the same time, we hold close Moss' (2019) comment that: "we should never underestimate the possibilities of bottom-up transgression, especially when driven by belief and desire" (p. 122). From our perspective, the "belief and desire" to which he refers are almost always an outcome of politics of refusal at the minor level. Insight makes way for passion and imagination, which makes way for new possibilities. But, it is cyclical in the sense that one also needs the opportunity, safety, and support (i.e., the conditions) to critically reflect on and reject the status quo. In our context, the conditions to support this work are not currently available to, or imaginable for, all educators and allies. We cannot imagine a way to get there without simultaneously engaging at the major and minor level. Care of the self—through a politics of refusal or other means—is unrealistic in conditions of ongoing scarcity, exploitation, and isolation. In fact, it may be dangerous for educators to care for themselves, if it defies the status quo and/or legal requirements of the job (Lisa's reflection below illustrates this).

We see a politics of refusal as necessary but not sufficient to both imagining possibilities and changing realities in early childhood education. In isolation or fragments, a politics of refusal creates barriers. It does not lead us to systems change and could potentially harm those who engage in it on their own. However, when ECEs collectively share the power of their refusals and/or the impact of their refusals and bridge the minor and major, we generate new discourses and the potential for transformative change. Once again, we

turn to feminist care ethics to guide when, how, and with whom one politically refuses and/or resists.

### 1.4. Caring with: Insights from a Feminist Ethics of Care

Feminist care ethics, developed by pioneering feminist scholars such as Nell Noddings, Virginia Held, Carol Gilligan, Eva Kittay, Selma Sevenhuijsen, and Joan Tronto, is not new to the ECE community. In a recent interview with Pacini-Ketchabaw and Moss (2020) draws on FCE to describe how ECE spaces are inherently political and ethical spaces. Supporting this assertion, he draws on FCE as a theoretical framework in relation to the minor level. He acknowledges the importance of phases of care (and their moral qualities) which include caring *about* (attentiveness), caring *for* (responsibility), care-giving (competence), and care-receiving (responsiveness).

Of particular interest to us, and not mentioned in this interview, is the recent shift to embracing FCE as political theory (see Engster and Hamington 2015; Tronto 2016; Sevenhuijsen 1998; Robinson 2011). This shift to thinking with FCE at the theoretical/political level brings attention to a fifth phase of caring: caring *with.* To care *with* others means that "care needs and the ways in which they are met need to be consistent with democratic commitments to justice, equality and freedom for all" (Tronto 2013, p. 23). In this way, we assert that transcending major and minor political spaces in relation to others in an intentional way is a critical component of ethical, political practice. It allows activism to be (re)conceptualized as an ethical caring action and experience. While Tronto's (2013) descriptions of caring *with* does not offer (nor do we think it should offer) a "how to" guide, it does suggest that reflecting on our own power and privilege and opening ourselves to other ways of being, thinking, and understanding with others is critical to democratic relations within and between major and minor spaces. If and how we have been cared *with* in our activism work, as well as reflecting on our attempts to care *with* others, is, thus, central to our analysis.

Tronto (2013) warns that "privileged irresponsibility", whereby any one person or group gets a "pass" out of care at any of the five phases of caring, is dangerous. As has been established by the Canadian feminist political economy and the care ethics literature, it is typically the most privileged in society (i.e., male and/or upper-class and/or White) who get "passes" out of caring at the expense of racialized, marginalized, classed women (Engster and Hamington 2015; Tronto 2013). On a local, national, and global level, women disproportionately fill the care void in increasingly private (home or market), unpaid or poorly paid, less-than-optimal working conditions (Arat-Koç 2006; Bezanson et al. 2015; Luxton 2006; Robinson 2011). We, thus, placed a concerted effort on naming times when we either participated in or observed "privileged irresponsibility" in our conversations and analysis.

It is important to address a key critique of FCE: that it centres the care experiences of White women to the occlusion of other, racialized groups. This critique has also been made of the women's and childcare movements. Crenshaw's (1989, 1991) pioneering work on intersectionality and Hill Collins' (2000) conceptualization of the matrix of domination, which emerged from the lived experiences of Black women, are also woven into our thinking with FCE. Crenshaw (1989) powerfully highlights how it is necessary to acknowledge and appreciate the intersections of oppressions to grasp the complexity of women's experiences in the world, claiming "any analysis that does not take intersectionality into account cannot sufficiently address the particular manner in which Black women are subordinated" (p. 140). Collins (2000) extends this thinking into the larger system of "a matrix of domination", making visible the "social organization within which intersecting oppressions originate, develop, and are contained" (p. 228). Taking an intersectional approach is critical to understanding women's unique identities and experiences as intersectional and situational and to resisting the urge to universalize the meaning of care and care work (Hankivsky 2014; Robinson 2019; Tronto 2013). As Perez (2017) suggests, "entanglements with matrices of

domination create social and institutional contexts where one experiences both power and oppression depending on her or his multiple and intersectional identities" (p. 53).

Because we are four privileged, White women in positions of leadership in the Ontario ECEC sector, and given that, in this sector, Black, Indigenous, and racialized educators often face persistent silencing and the most precarious working conditions and wages, we are frequently reflecting on how our privilege may cloud our understandings and how our activism must be informed by multiple experiences/knowledges—particularly those experiencing multiple oppressions from racist, colonial, and neoliberal systems. Further, it is precisely because of this that we feel it is necessary to both name our privilege and consistently reaffirm our commitments to asking questions, listening, and caring *with* others who are different than us in our activism. Interestingly, a "caring activism" is a concept used by Black scholars to explain the actions of Black leaders in the de-segregation in education movement in the United States. Ramsey (2012) argues that "Black womanist teachers recognized and learned from traditions of female activism, [and] embraced caring as a key force for social activism" (p. 245). Her overall argument fits well with where we land: caring *is* activism.

## 2. Method

This project, and paper, has truly evolved over time. There have been several iterations, conference presentations, and manuscripts thrown out and picked back up again. The four authors initially met in the spring of 2021 on a hunch that we had to think more deeply about what activism means and/or what it looks/feels like in our sector. We agreed to have regular meetings, at first monthly and then more sporadically (often in relation to conference presentations or manuscript preparations). Throughout these conversations we came to identify key focusing events or experiences in our own activism journeys, which we then more formally wrote up as narrative accounts.

Though not intentional at the time, we now recognize we embraced an autoethnographical approach to inquiry, whereby we "wrote about epiphanies that stemmed from, or were made possible by, being part of a culture and/or by possessing a particular cultural identity" (Ellis et al. 2011, p. 275). This methodological approach, with its deep commitment to "research as a political, socially-just and socially-conscious act" (Ellis et al. 2011, p. 273) gave us space to understand ourselves and our experiences as early childhood education activists in relation to each other and the broader context in which we have been, and continue to be, embedded. We discovered that our relationships with each other, and others, were central to the personal and professional paths we have chosen. For example, when difficult topics, memories, or emotions emerged in writing and sharing our narratives, we held space for that person both in that moment and in the moments, days, and weeks to follow. All, or a couple of us, engaged in several follow-up "check-in" conversations, where we were able to both give and get feedback about how we felt cared for—or not. These conversations were a catalyst for us to understand our lives, journeys, and motivations differently (and sometimes painfully). We both laughed and cried, but truly came to appreciate how foundational care has been and continues to be in engaging in/sustaining our activist work. Out of this process, we imagined the kind of activism we want to engage with in the future.

## 3. Findings and Discussion

Thinking with feminist care ethics, we noticed several themes in our narratives and discussions: (1) Being cared *for* is a necessary component of caring *with*; (2) The politics of refusal can be privileged irresponsibility; and (3) Activism requires a repositioning of individual pathology to structural impossibilities. Overall, we observe that being sensible about change is radical and being radical is sensible in our contemporary ECE context in Ontario. Below, we elaborate more thoroughly on these themes and how they inform a reconceptualization of activism.

### 3.1. Care Begetting Care

In reflecting on our narratives, we came to appreciate how our relationships with each other are what keeps us motivated to critically engage with ECE policy and practice. Rather than pursuing the fictitious ideal of being that the independent individual neoliberalism romanticizes and the existing social and political structures rewards, we all agree that feeling cared for by others in our personal and professional lives (two worlds that are not qualifiedly distinct for any of us) has offered us a more meaningful existence and source of inspiration and motivation for our activism. This observation is consistent with Almeida (2019), who asserts that "prolonged participation (in social movements) requires deep social ties to other activists" (p. 116). We feel we could not engage in and sustain this difficult work without each other's continued support, as there are often few (if any) concrete practice and/or policy "wins". Yet, we know that is what makes it all the more necessary.

While all four authors echoed the centrality of feeling cared *for* to care *about* and *for* others, Brooke shared an example that stood out. She shared how feeling cared *for* by her colleagues during a personal crisis was pivotal to understanding her potential as an activist. Early in her career, she found herself a single mother escaping domestic violence. Having grown used to the experience of being ignored, silenced, and gaslighted, her peers' and professor's care for her and her baby when she made the decision to leave (and stay away) opened up an entirely new way of being in the world—one that included her voice. Upholding Tronto's five phases of care, Brooke felt noticed (cared *about*) and supported (cared *for* in that her peers held her baby during class time and while she caught up on reading/assignments in the student lounge). Most of all, she felt heard/seen in that what she needed was being taken seriously (care-*receiving*). It was because she felt cared for (i.e., her experience mattered to people) that she had the courage to engage in activism through sharing her story at a public childcare consultation (caring *with*). To her surprise, her story was of great interest to the panel, and they asked her several questions. That was a pivotal moment for Brooke; she came to understand she had a voice, that her story was worth listening to, and that *all* people deserve to be seen and heard. There have been hundreds of moments, and some truly harrowing moments, since then where Brooke has felt supported and cared for by her colleagues—fellow advocates, educators, and researchers. We all agree that our work would not, and could not, be sustainable, without us first being cared *about* and *for*.

### 3.2. Privileged Irresponsibility, Intersectionality, and the Dangers of the Politics of Refusal

We see moments and experiences of discomfort as necessary for transformative change in ECE. For us, engaging the voices and lived experiences of ECEs is necessary to resist the binary between major and minor spaces and destabilize the status quo. We recognize it is also a significant departure from traditional advocacy work, whereby ECEs engaging politically may be seen as ontologically conflicting with professionalism (Macdonald et al. 2015). When we include ECE voices and honour their different perspectives, we encounter constant tensions. One of the key tensions relates to those who refuse to take on the ethical responsibility of care, especially in this context of caring *with*. Activists must be deeply aware of their own privilege and understand multiple sources of oppression (e.g., racism, heteronormativity, ableism, ageism, and classism) in minor and major spaces and the ways in which these intersecting oppressions produce different kinds of power relations and social inequities (Collins 2000; Crenshaw 1991). They (we) must also notice that activism, when not conducted with care, has the potential to disempower, silence, and cause harm. This is particularly important as we respond to our racist, neoliberal, colonial, and patriarchal context and requires that we listen to, value, and centre the lived experiences/truths of educators. We believe that intersectional activism through caring *with* is necessary to disrupt and shift power relations in the matrix of domination and create meaningful movement between minor and major spaces.

We are concerned that the politics of refusal holds potential to be a "pass" out of *caring about* and *caring for*. We are concerned that enacting a politics of refusal, confined to the minor level, may cut off opportunities to disrupt, negating the possibility of moving between major and minor political spaces and, therefore, failing to interrupt existing systems/power relations. If we refuse the major level and/or do not/cannot think about our refusal as a broader political statement, we risk placing the responsibility to change oppressive systems solely on the oppressed, who at times lack real opportunities to "refuse" without consequence. Simultaneously, we recognize that remaining active only at the major level causes harm by excluding and oppressing the voices, experiences, and knowledges of those living, working, and caring in early childhood spaces. There must be intentional movement between these spaces.

We are not suggesting that all activists must instigate resistance—especially not alone. As pointed out by other activism scholars (Almeida 2019; Kahn and Lynch-Biniek 2022), there are very real barriers and/or "risks" to resisting—particularly for marginalized groups. But, this is even more reason why those of us in positions of power within the matrix of domination (Collins 2000) have a responsibility to *care for* (be attentive to) and *care about* (take responsibility for) others (Tronto 2013). To do this, we must center the intersectional experiences of early childhood educators, particularly those who are most often silenced (Crenshaw 1991). Those of us who safely can *must* actively disrupt the hegemonic status quo at both major and minor levels. We cannot ensure ECEs are listened to or cared for at the major level, but we can ensure our activism is caring *with* them.

When reflecting on privileged irresponsibility, intersectionality, and the politics of refusal, Alana shared her discomfort with the "devastatingly impossible task" of representing and/or bringing ECEs into the policy-making arena. As the Executive Director of a key ECE organization in Ontario, Alana is objectively positioned as a gatekeeper, something inevitable when policy processes are so far divorced from the lived experiences of educators. Alana deeply appreciates the contributions ECEs make to policy debate and dialogue and described how she feels a weighty responsibility to use her access to spaces of power to disrupt the status quo and push for caring possibilities. But, at times, she described feeling hopeless and immobilized in major political spaces—especially in our current Ontario context, where care as a value and practice is actively disregarded. At the same time, she is continually hearing from ECEs about the impossibilities in their minor spaces, where the pandemic has exacerbated pre-existing challenges, the gap between the pedagogical and/or decent work 'haves' and 'have nots' is growing, racism is silenced, and educators experiences and knowledges are never "good enough" for those in decision/policy making positions. Their continued hope in this context is the undeniable reason why she continues the incredibly trying work of creating points of intersection between major and minor political spaces.

In thinking more about Alana's privileged position as a gatekeeper, something to which all authors could relate, she proposed reconceptualizing the gate at a dam. Where the formal systems/governments are the physical structures that make up the dam, they open to allow certain things through (some governments are more interested in opening these more frequently to a wider body of stakeholders than others). As the ED of a respected ECE organization in Ontario, she has had the privilege of steering a boat that is usually allowed through the dam (although almost always under conditions we do not determine). Her boat is comprises the voices and lived experiences of ECEs with whom we are continually in contact. While she cannot bring all ECEs on her boat (though she has been known to have a stowaway here and there), she makes every effort to ensure ECE truths, needs, and desires get through that dam.

And dams are not perfect. Stones, sediment, fish, or even a stray kayak may also make their way through the dam when it opens. The dam itself cannot control what comes through once it is open. Through Alana's reflection and boat and dam imagery, we came to understand that our role as activists is to let as much through as we can while the gate is open. And once we are through, we do everything we can to ensure that we are heard.

Of course, we cannot control what is heard or ignored, but we can live with ourselves knowing we are doing what we can to maximize every opportunity to centre the voices of ECEs when that dam opens. In our ongoing work, we continue to strategize how we can maximize the responsibility that comes along with our privilege.

*3.3. Shifting from Individual Pathologizing to Structural Impossibilities*

Throughout our conversations, we became aware of an intense sense of individual responsibility for a chronically under-resourced and undervalued profession. We recognize that a collective sense of responsibility toward children, families, and educators, rooted in our deep commitments to them, is what brings us to our work in ECE. We stand with ECE scholars who have long criticized neoliberalism's (convenient) hyper-individualization of social problems so that public and private institutions can avoid caring about, for, and with others (Engster and Hamington 2015; Slote 2015). It is also what incites a refusal of neoliberalism's individualizing and weaponizing of our beliefs and desires against us. When we cannot act in a way that reflects our deep commitments to our communities (for example, being unable to sit one-on-one with a distressed child due to not having enough time or staff) or when we do act in a way that reflects our commitments but it violates standardized measures (typically provincial regulations as in the Lisa's story below), *we* feel inadequate and/or that we, as individual humans, are letting people down.

Capturing this, Lisa shared her story of being reprimanded and feeling ostracized when acting in a way that upheld her ethical values and beliefs in relation to her work with children and families. Working in a lab school childcare program for 15 years, Lisa was grateful to have a job that paid well and offered benefits. What she not have was paid planning time. Living through the professionalization of the ECE sector in Ontario, she remembered feeling overwhelmed and increasingly time-strapped by the mounting regulatory demands on ECEs that took her away from the children. In her own words, "the increased paperwork related to curriculum and planning, externally imposed and narrowly defined professional development requirements and other seemingly meaningless tasks took the 'joy' out of her work". One summer, she made the explicit decision to focus her energy, time, and attention on children and families and let the paperwork wait. This plan backfired when a licensing inspector found her program plan incomplete. She was positioned as, and felt like, a "bad" educator, carrying the shame of this with her for years. The structural impossibilities of Lisa's situation (24 children in a preschool room, no planning time, and an endless pile of paperwork) laid the blame for the licensing transgression squarely on her shoulders.

In different ways, we could all relate. At some point in all our careers, we all felt the structural impossibilities in which we were embedded were our problems to fix (or at least compensate for) as individuals. And if we could not fix it, we were not committed enough to our work and, therefore, not good enough educators. Our internalized narrative was that we needed to be better, more organized, more time efficient, smarter, etc. Lisa refused this narrative and the broader hegemonic politic which it feeds. With support from fellow ECE activists, Lisa was able understand her situation differently—one of systemic failings rather than a personal deficiency.

Acting on this at the minor level, Lisa engaged in "bottom up transgression" (Moss 2019), speaking out against a lack of paid planning time in her program. While it felt risky for her to speak up in this way, and it came with consequences (being reprimanded by her employer and a formal notice of regulatory violations), she was still protected from more severe consequences by her unionized position, a position that came relatively easy to her as White educator. This experience also prompted her to take a leadership role in forming a community-based group of ECEs that advocated *with* and *for* each other. Several years later, as a Master's of Arts student in Early Childhood Studies, critical reflections and conversations Lisa had with Rachel (her professor at the time) allowed her to identify her actions as "politics of refusal". Far from being a pass out of care, Lisa acted to disrupt the

power of major politics over ECEs in minor spaces in a way that embodied caring *about* and *for* herself, her colleagues, children, and families.

*3.4. Childcare Activism: When the Sensible Is Radical*

Often activists distinguish between sensible reforms (advocating for incremental policy changes) and radical activism (e.g., seeking a complete overhaul of the ECE system). These terms could be read in two ways. First, they could represent a type of binary model in which sensibleness is pitted against radicalness. Sensible advocacy maintains that efforts that push for comprehensive system level changes can alienate governments that prefer step-by-step changes to existing policies rather than transformative policies. Radical activists often work in overt opposition to governments, being transparent in their short- and long-term goals of system overhaul.

A second way to read the definitions above is that something sensible can also be radical. At one point in her narrative, Alana said "our efforts to move between the major and minor political spaces through centring the voices of educators in policy discussions/directions is too radical for some". Reflecting further on this, we think that it is *im*possible in ECE *not* to be radical if one is sensible enough to prioritize the voices and experiences of early childhood educators. The gap between the professional status of Ontario ECEs and the complexity of their work *and* their poor wages and working conditions is so radically incongruent that the only sensible thing to do is enact sweeping change, such as immediately implementing a wage grid starting at CAD 30/h for qualified educators.

In her role of director of a post-secondary Early Childhood Studies (ECS) program, Rachel shared how she used every opportunity out of a "volatile combination of frustration and desperation to assert an image of early childhood educators as intelligent, competent, and powerful". Doing this was, and continues to be, the epitome of both sensible and radical. The value and efficacy of ECEs is continually undermined by almost everyone—not just politicians and policy-makers but sometimes those in positions of power within the sector itself. Just like conceptualizing educators as capable, intelligent, competent, and powerful is radical, the idea of a publicly funded wage grid starting at CAD 30/hour is (sadly) seen as radical to many involved in creating and/or advocating for childcare policy. But, it truly is the most sensible solution to the chronic undervaluing of caring for/with children and families. When what is necessary and what exists are so far apart from each other, the radical is sensible and the sensible is radical. Circling back to FCE, this assertion reflects Tronto's (2013) broader sensible/radical philosophical argument that it is necessary to organize society in a way that meets care needs *before* profits—rather than the other way around.

Naming the fallacy of the sensible/radical binary in relation to childcare activism also offers an answer to the political strategy/shapeshifting issues mentioned in the introduction. When the sensible *is* radical, incremental policy changes, and/or approaches to activism that remain defined within the parameters of neoliberal thinking (i.e., a human capital discourse) does little more than tinker at the edges of a market-based system. A transformational activism requires we be authentic to ourselves and each other by acknowledging what is possible where we are *and* continuing to think outside of what already exists. Such imaginings can be accomplished only in the context of supportive, responsive caring relationships.

## 4. Conclusions

Coming full circle, our experiences and efforts towards reconceptualizing activism in ECE are well summed up by the quote included at the beginning of this article: "bringing the question of woman into the philosophical canon essentially deconstructs that cannon. Women don't fit" (Hekman 2014, p. 24). ECEs, who are overwhelmingly women, will never fit in the formal patriarchal, political structures that currently exist. What we can do is reconceptualize how we *think about* and *do* politics in the hopes that the cannon can

be melded into something different. Rather than our shapeshifting or watering down our message/goals, we have to reshape the cannon. We must create a place for us.

We have made the argument that a caring activism requires movement within and between major and minor political spaces. Yet, we recognize such movement cannot be accomplished alone. Through reflecting on our experiences, we have come to deeply appreciate how important it has been for us, as individuals, to feel cared *about* and *for* in finding the courage and strength to take on the broader system issues. Indeed, we wonder if we ever would have come to understand our struggles as endemic to a hegemonic political order, rather than personal shortcomings, without the networks, communities, and relationships we have created with each other and our colleagues.

We cannot, and do not need to, do this work alone. It is our assertion here that adhering to the principles of feminist care ethics—caring *about*, *for*, and *with* others—is necessary to get the irons hot enough to melt the cannon down and start reshaping. Our ongoing relationships with each other and ECEs, and an ongoing humility to remain open to whatever is next, allow us to continually understand our world differently and act with others in sensible, radical ways. We enthusiastically invite others to join us in thinking and carrying out ECE activism differently, exploring and navigating the beautiful awkwardness, discomfort, tension, and possibilities in the space between the major and minor and in caring for and with each other.

**Author Contributions:** Conceptualization, R.L., B.R., A.P., L.J.; methodology, B.R., R.L.; software, none; validation, B.R., A.P., L.J., R.L.; formal analysis, B.R., A.P., L.J., R.L.; investigation, B.R., A.P., L.J., R.L.; resources, B.R.; data curation, B.R., A.P., R.L., L.J.; writing—original draft preparation, B.R.; writing—review and editing, B.R.; visualization, B.R. and R.L.; supervision, R.L. and B.R.; project administration, B.R.; funding acquisition, none. All authors have read and agreed to the published version of the manuscript.

**Funding:** This research received no external funding.

**Institutional Review Board Statement:** This theoretical paper did not require ethics approval.

**Informed Consent Statement:** Not applicable.

**Data Availability Statement:** Data sharing not applicable.

**Conflicts of Interest:** The authors declare no conflict of interest.

## Note

1    The one clear exception is the province of Quebec, which embraces a more European approach to childcare provision. Since 1997, reduced fee contribution childcare spaces have been funded by the provincial government, whereby parent fees have remained significantly lower than the rest of Canada.

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
