# Peer review of "Reconceptualizing Activism through a Feminist Care Ethics in the Ontario (Canada) Early Childhood Education Context: Enacting Caring Activism"

_socsci, doi:10.3390/socsci12020089_

Round 1
Reviewer 1 Report
This paper was a pleasure to read. It is well structured, conceptualised and written. The paper discusses an important topic relevant to ECE worldwide. The writing is both appropriately academic and personal – an approach I much appreciate. To start, the authors have made clear their positioning as educators and researchers, provided a detailed context and explanation of key terminology, and offer a useful critique of FCE. The findings and discussion tie together well, providing useful suggestions for practical application in our work as educators and activists. I wonder if the findings section should actually be called ‘findings and discussion’ and the discussion should be ‘conclusion’ – a minor point for the authors to think about. A small number of additional suggestions are in the table below. Overall, an important and robust article well worth publishing.
|
Page 1, line 20 |
Word ‘to’ missing after ‘necessary’ |
|
Page 2, line 62 |
AECEO needs to be written in full |
|
Page 3, line 112-113 |
I would recommend moving the examples of major political spaces to before the sentence about these spaces being high stakes. Possibly also move the high stakes sentence to after the sentence about these being exclusionary to those without power. |
|
Page 3, line 194 |
This is the first time FCE is used in the article. The acronym is introduced in the abstract but not in the article itself. Needs to be introduced before it is used. |
|
Page 8, line 397 |
Full stop missing |
Reviewer 2 Report
kia ora from down under
Thank you for the opportunity to review this paper. It reads well and clearly makes the point at a time when there is much happening in the Canadian context.
The use of FCE to provide a framework for analysis and then to broaden its power by adding caring with to address the minor and major political spheres of activism is innovative in the theoretical context.
As part of my own advocacy I would suggest adding ableism to your list intersectional oppressions on p.7.
A question - the title, abstract and keywords makes no reference to Canada. In an international context will people find their way to your article via Ontario?
In the document returned I have highlighted minor typos. The first one is a missing reference for Hekman. Great to see a return to the quote in the discussion.
thanks again
